# The Use and Effectiveness of Selected Alternative Markers for Insulin Sensitivity and Secretion Compared with Gold Standard Markers in Dietary Intervention Studies in Individuals without Diabetes: Results of a Systematic Review

**DOI:** 10.3390/nu14102036

**Published:** 2022-05-12

**Authors:** Lucia Vazquez Rocha, Ian Macdonald, Marjan Alssema, Kristine Færch

**Affiliations:** 1School of Biosciences, University of Nottingham, Loughborough LE12 5RD, UK; 2School of Life Sciences, University of Nottingham, Nottingham NG7 2UH, UK; ian.macdonald1@nottingham.ac.uk; 3Nestle Institute of Health Sciences, 1015 Lausanne, Switzerland; 4Unilever Research and Development, 3133 AT Vlaardingen, The Netherlands; marjanalssema@hotmail.com; 5Steno Diabetes Center Copenhagen, 2730 Herlev, Denmark; kristine.faerch@regionh.dk; 6Department of Biomedical Sciences, University of Copenhagen, 2200 Copenhagen, Denmark

**Keywords:** insulin sensitivity, insulin secretion, gold standard, surrogate markers, dietary intervention studies

## Abstract

Background: The gold-standard techniques for measuring insulin sensitivity and secretion are well established. However, they may be perceived as invasive and expensive for use in dietary intervention studies. Thus, surrogate markers have been proposed as alternative markers for insulin sensitivity and secretion. This systematic review aimed to identify markers of insulin sensitivity and secretion in response to dietary intervention and assess their suitability as surrogates for the gold-standard methodology. Methods: Three databases, PubMed, Scopus, and Cochrane were searched, intervention studies and randomised controlled trials reporting data on dietary intake, a gold standard of analysis of insulin sensitivity (either euglycaemic-hyperinsulinaemic clamp or intravenous glucose tolerance test and secretion (acute insulin response to glucose), as well as surrogate markers for insulin sensitivity (either fasting insulin, area under the curve oral glucose tolerance tests and HOMA-IR) and insulin secretion (disposition index), were selected. Results: We identified thirty-five studies that were eligible for inclusion. We found insufficient evidence to predict insulin sensitivity and secretion with surrogate markers when compared to gold standards in nutritional intervention studies. Conclusions: Future research is needed to investigate if surrogate measures of insulin sensitivity and secretion can be repeatable and reproducible in the same way as gold standards.

## 1. Introduction

Type 2 Diabetes (T2D) is a chronic, lifestyle-related disease characterised by elevated circulating glucose levels. The International Diabetes Federation (IDF) has estimated that 624 million adults, aged 20–79 years old, will have T2D by 2040 [1]. Currently, T2D comprises 90% of all diabetes cases worldwide and is a major cause of blindness, kidney failure, heart attacks, stroke, and lower limb amputations [2].

Insulin resistance and diminished insulin secretion are key characteristics of the progression to type 2 diabetes [3], and are often used as outcomes in intervention trials aimed at preventing the development of type 2 diabetes. The assessment of insulin sensitivity and secretion is vital for the early identification of people at risk of developing diabetes as well as for implementing interventions to prevent or delay the pathological progression of the disease.

The classical test to assess insulin sensitivity is the euglycaemic-hyperinsulinaemic clamp (EHC) [4]. During the EHC, insulin concentration is elevated to a standardised level by infusing insulin at a constant rate for 2–3 h. Hyperinsulinaemia stimulates glucose utilisation and suppresses glucose production. Glucose concentration is kept constant at a fixed value (typically ~5 mmol/L) utilising a variable glucose infusion. The variable glucose infusion is adjusted based on frequent glucose measurements, according to an algorithm. Insulin sensitivity is determined from the mean glucose infusion rate at the end of the test when the glucose infusion rate has stabilised (the steady-state period). Under these conditions, the rate of glucose infusion is nearly equal to the rate of glucose utilisation at the standardised insulin and glucose levels, as glucose production is strongly suppressed [5]. Therefore, the rate of glucose infusion, the simplest index of the glucose clamp, is approximately equal to the rate of glucose utilisation.

Another assessment of insulin sensitivity is the intravenous glucose tolerance test (IVGTT) [6]. In this test, glucose is injected intravenously (typically 300 mg/kg), and glucose and insulin concentrations are measured for 3–4 h. The time course of glucose and insulin is analysed with a simplified model of glucose metabolism (the minimal model) to obtain an index of insulin sensitivity denoted as S_I_. In an important variant of the protocol [7,8,9] a bolus injection of insulin (180–300 pmol/kg) or a brief insulin infusion, given 20 min after glucose injection is performed to increase insulin concentration. The IVGTT is experimentally simpler than the EHC. Data analysis, however, requires specialised software and expertise [10].

For insulin secretion, defining a reference method is more problematic than for insulin sensitivity. Standardisation of the glucose levels is only possible with intravenous glucose infusion.

The reference methods with intravenous glucose infusion are the hyperglycaemic clamp (HC) and the first phase insulin secretion of IVGTT or acute insulin response to glucose (AIRg). IVGTT also provides an estimate of insulin sensitivity, as detailed previously.

AIRg is characterised by an abrupt rise in plasma insulin levels over three to five minutes followed by a decline. The classical index of first-phase insulin secretion is the incremental area under the insulin or C-peptide curve in the first eight-ten minutes. AIRg is a good predictor of diabetes [11].

The main problem with EHC, IVGTT, and AIRg is that, even though they are considered gold standard techniques, they may be perceived as invasive, impractical, and expensive for use in dietary intervention studies. All these methods imply problems for the patients. Therefore, surrogate markers provide indirect estimates; for example, insulin (or C-peptide) and glucose measurements in blood samples taken in the fasting state or after a glucose challenge or a standardised meal.

Surrogate markers for insulin sensitivity and secretion are based on established principles of glucose kinetics and insulin-mediated glucose disposal, and most have been developed from or validated against the euglycaemic-hyperinsulinaemic clamp [12].

Surrogate measures are divided into two families: the oral glucose tolerance test (OGTT) indices and the fasting indices. The OGTT’s surrogate measures are e.g., the Stumvoll Index and the Matsuda Index. During OGTT, blood glucose and insulin levels are assessed in a fasted state (t = 0) and usually at three different time slots (t = 30, 60, and 120 min) after the consumption of a standard oral glucose load of 75 g [13]. The fasting surrogate measures include the homeostatic model assessment for insulin resistance (HOMA-IR), the quantitative insulin sensitivity check index (QUICKI), and fasting insulin.

HOMA-IR was originally derived from a mathematical model of glucose homeostasis [14], as the product of fasting glucose and insulin and a scaling factor to yield a value of 100% in individuals with normal insulin sensitivity. QUICKI is calculated from fasting glucose and insulin concentrations and is mathematically equivalent to the inverse of the logarithm of HOMA-IR plus a constant [15]. Fasting insulin has been considered the most feasible way of measuring insulin resistance [16]; however, one main disadvantage of the method is the lack of accuracy or precision of available insulin assays [17].

Surrogate markers of AIRg are HOMA-beta and the disposition index (DI)—the latter taking the level of insulin sensitivity into account. The analysis of the relationship between insulin secretion parameters and insulin sensitivity has led to the concept that the two variables are hyperbolically related [18]. The product of the insulin secretion and insulin sensitivity parameters, which is constant along the hyperbola representing the normal relationship, has been proposed as an index of beta-cell function properly corrected for insulin sensitivity. This index has been denoted as the “Disposition Index”.

The advantage of surrogate markers is the less complex and invasive nature of the participant phenotyping. There have been several highly cited and extensive reviews in the past on surrogate markers of insulin sensitivity [19,20,21,22] and insulin secretion [23,24,25,26]. The overall conclusion of the reviews was that surrogate markers are useful for large epidemiological studies and that they are appropriate when the investigators are aware of their limitations.

The novelty and importance of this systematic review focus on the use of surrogate markers in nutritional intervention studies, when compared with gold standards, which have not been previously reviewed. Moreover, the use of these measures among individuals without diabetes has not been extensively reviewed. This is of importance because of the need for sensitive markers to evaluate the efficacy of trials aiming to prevent the development of diabetes. The present review investigates surrogate markers of insulin sensitivity and secretion compared to gold-standard methods, in response to a dietary intervention, which will aid the design of future whole dietary patterns, food and food bioactive interventions studies that target insulin sensitivity and secretion.

## 2. Materials and Methods

### 2.1. Search Strategy and Selection Criteria

#### 2.1.1. Inclusion Criteria

Eligible studies were English-language reports of all dietary intervention trials and their influence on insulin sensitivity and secretion markers as evaluated by both a gold standard method and a surrogate marker.

For insulin sensitivity, the gold standard methods were the EHC (M-value unit) and IVGTT (S_i_ unit). For insulin secretion, AIRg was used as the gold standard. The surrogate markers of insulin sensitivity were either HOMA-IR, fasting insulin, or the area under the curve oral glucose tolerance test (AUC OGTT). For insulin secretion, the disposition index (DI) was used as a surrogate marker.

Studies were limited to those that included only nondiabetic populations with no restriction on age. However, participants could have treated hypercholesterolemia, treated hypertension, had prediabetes, or been obese if they were considered otherwise healthy. The search strategy used can be seen on Table 1.

#### 2.1.2. Information Sources

Searches were performed between the 18 of February 2016 and the 21 of June 2017. We systematically searched for title articles in PubMed, Scopus, and Cochrane databases.

#### 2.1.3. Selection Process

The titles and abstracts of studies identified by the search were screened by two reviewers and ineligible studies were excluded. The abstracts of all potentially relevant studies were obtained and assessed independently for eligibility by two independent reviewers who completed an abstract review form. Any disagreement was resolved by discussion.

### 2.2. Data Items

#### 2.2.1. Types of Studies

Randomised controlled trials, including crossover and parallel arm studies, and longitudinal randomised studies were included for this review. PRISMA guidelines were used to conduct the study and report this project.

#### 2.2.2. Types of Intervention

“Dietary interventions” refer to all interventions with either change in total diet or one or more specific foods or dietary components. Exotic plants were not considered in this review as they can be seen as pharmacological rather than nutritional interventions. Studies measuring meal effects such as challenge tests were also excluded. The dietary intervention of the studies had to be at least 1 week in duration and included studies of a combined dietary and physical activity intervention. All included interventions had a control group.

### 2.3. Data Extraction and Quality Assessment

Relevant data were extracted by the lead author and subsequently reviewed by one of the co-authors for accuracy. The information extracted from the surrogate and reference markers of insulin sensitivity and secretion were measurement units, number of intervention arms, number of control arms, baseline measurements, end of study measurements, and change mean.

The quality of the studies was evaluated using a risk of bias assessment (Cochrane) with criteria that included the following: random sequence generation, allocation concealment, blinding of participants and personnel, incomplete outcome data, selective reporting, and other sources of bias.

### 2.4. Data Synthesis and Analysis

Correlation is the most popular method used to compare the efficacy of surrogate markers with gold standards as it can give information regarding the power of a relationship between two variables [27]. However, either Spearman or Pearson correlations rely on the assumption of a bivariate normal distribution.

Thus, Lin’s concordance correlation coefficient (LCCC) has been proposed as an analysis method. Lin measures bivariate pairs of observations relative to a “gold standard” test or measurement. Lin calculates the degree to which two sets of data descend along a line of x = y when plotted as opposed to another [28].

LCCC is analogous to Bland and Altman’s concept of the line of the agreement [29]. This concept is especially useful when two variables are measured on different scales. Consequently, Bland and Altman proposed that the extent of agreement could be investigated by plotting the differences between the pairs of measurements on the vertical axis, against the mean of each pair on the horizontal axis.

To compare studies with different units of measurement, the responses were expressed as % change from the baseline/starting value. The percentage change (PC) in response to each treatment, either using the surrogate or gold standard method, was calculated as (Value–Value1)/Value1) × 100) for both the intervention and the control group. This equation quantified the change from one value to another and expressed the change as an increase or decrease.

Once the proportional changes were obtained, we compared EHC (M-value), IVGTT (S_i_), and AIRg with each surrogate marker using Lin’s concordance correlation coefficient, with 95% confidence intervals. The correlations were conducted on the mean values from each trial rather than individual participant data. Data were analysed using Stata 17 software (StataCorp. 2021. Stata Statistical Software: Release 17. College Station, TX, USA: StataCorp LLC).

For interpreting Lin’s concordance correlation coefficient results, McBride’s descriptive scale for values was used [30]. An agreement was defined as a concordance correlation coefficient of >0.99.

Data were assessed for normality using the Shapiro–Wilk test, which revealed non-normal distributions. Robvis tool was used to present the results of the risk of bias assessment [31] (Appendix A).

## 3. Results

A total of thirty-five articles met the inclusion criteria. A PRISMA flow chart describing the selection process is represented in Figure 1.

From the represented articles, a total of 2047 individuals (aged between 15 to 75 years of age) were involved in the primary analyses. Each article provided two study arms: control and intervention. Only the intervention arm data was used.

Among the thirty-five articles, the majority were conducted in Europe (twenty-one), four from Australia and New Zealand, and ten from North America. The studied population was mainly adults (including young, middle-aged, and older ones). Two articles were performed on Latino and minority adolescents.

The basic details of each article, as well as the characteristics of the study participants included in this systematic review, are presented in Table 2 and Table 3.

Insulin sensitivity levels were significantly modified after the nutritional intervention in three studies measured using the EHC and the surrogate marker of fasting insulin [32,33,34] as well as AUC OGTT and fasting insulin [35], Appendix A (*p* ≤ 0.05).

Using IVGTT and fasting insulin, five studies demonstrated a change in insulin sensitivity as compared with baseline [36,37,38,39,40], Appendix A (*p* ≤ 0.05). No change in insulin secretion was detected after the nutritional treatment in AIRg and DI, Appendix A (*p* > 0.05).

The detailed results of the risk of bias assessment using the Robvis tool are presented in the Appendix A. Nonetheless, twenty-four out of the thirty-five studies showed no information regarding allocation concealment and random sequence generation.

Analysis of the results of proportional change with LCCC, using the difference between baseline data and after the nutritional intervention of all studies, established a lack of agreement between the gold standards and the surrogate markers (<0.99) Table 4.

Figure 2, Figure 3 and Figure 4 show the visualisation of Bland and Altman’s plot relationship between paired (gold standards and surrogate markers) differences and their average. We found no agreement between the markers.

**Table 2 nutrients-14-02036-t002:** Euglycaemic hyperinsulinaemic clamp trials description.

Study	Study Design	Country	Subjects Characteristics	Dietary Treatment	Dietary Control	Dose	Sample Size	Duration (Days)	Funding Source
Bogdanski, 2013 [32]	Randomised, double-blind placebo-controlled study	Poland	Obese adults	L-arginine	Placebo	9 g	88 males and females	180	The Ministry of Science and Higher Education, Poland
Chachay, 2014 [41]	Randomised controlled trial	Australia	Healthy subjects (men)	Resveratrol	Placebo	3000 milligrams	20	56	Princess Alexandra Research Foundation, the Lions Medical Research Foundation, and the National Health and Medical Research Council of Australia
Derosa, 2012 [42]	Randomised, double-blind, controlled study	Italy	Adults with dyslipidaemia	Supplementation with n-3 PUFAs. The diet included 50% calories from carbohydrates, 305 from fat (6% saturated), and 20% from proteins, with a maximum cholesterol content of 300 mg/day and 35 g/day of fibre.	Placebo	1200 mg of EPA and 1350 mg of DHA	167 (82 males and 85 females)	180	*Not disclosed*
Grimnes, 2011 [43]	Randomised double-blind controlled trial	Norway	Healthy adults	Vitamin D	Placebo	20,000 IU	94 males and females	180	Norwegian Council of Cardiovascular Disease
Hays, 2006 [44]	Randomised controlled trial	United States of America	Elderly adults with impaired glucose tolerance	Low-fat diet and aerobic exercise	Control diet (41%fat, 45%carbohydrate and 14% protein)	Low-fat diet (18% fat, 63% carbohydrate and 19% protein). Aerobic exercise = 4 d/w, 45 min, 75–80% peak heart rate.	31 (18 females and 13 males)	84	National Institutes of Health grants
Hokayem, 2013 [45]	Randomised double-blind controlled trial	France	First-degree relatives of type 2 diabetic patients	Grape polyphenols	Placebo	333.33 mg grape extract/per capsule were taken daily. Three during breakfast and three at dinner	38 men and women	63	French National Research Agency
Johnston, 2010 [46]	Single-blind, randomised, parallel nutritional intervention	United Kingdom	Healthy subjects (women)	Resistant starch	Placebo	40 g per day	20	84	The National Starch LLC and by infrastructure funding support from the Medical Research Council and the NIHR Biomedical Facility.
McAuley, 2002 [47]	Randomised controlled trial	New Zealand	Healthy men and women	Change in diet	Control group was advised to continue their usual diet and exercise during the 4-month experimental period	Modest diet (M): <32% fat, 11% saturated fat, 14% monounsaturated fat, 7% polyunsaturated fat, 50% CHO, 18% protein, cholesterol targets < 200 mg per day and dietary fibre > 25 g per day. Intensive diet (I): <26% fat, <6% saturated fat, 13% monounsaturated fat, 7% polyunsaturated fat, 55% of CHO, 18% protein, cholesterol < 140 mg/day and dietary fibre > 35 g/day	77	120	The Health Research Council, Otago University, and the Otago Diabetes Research Trust, New Zealand.
Sanchez, 1997 [33]	Randomised controlled trial	Spain	Hypertensive patients	Calcium supplementation	Placebo	1500 mg/day	20 (12 men and 8 women)	84	*Not disclosed*
Tardy, 2009 [48]	Randomised controlled trial	France	Overweight women	Low-trans fatty acids (TFA) diet and ruminant trans-fatty-acids-rich lipids diet.	Industrial Trans Fatty Acid-rich lipids food (5.58 g/d)	Low-TFA lipids/d (0.54 g/d), ruminant TFA–rich lipids (4.86 g/d)	58	28	*Not disclosed*
Lagerpusch, 2013a [49]	Controlled, parallel-group feeding trial	Germany	Healthy men	Modification of dietary content	During control diet, the 50%CHO group was assigned to a low glycaemic (LGI) diet, while the 65% CHO group was assigned to a high glycaemic diet (HGI).	Participants were divided into two groups differing in macronutrient composition of the diet (50%CHO group: 50% CHO, 35% fat, 15% protein; 65%CHO group: 65% CHO, 20% fat, 15%protein). During refeeding, the 50%CHO and 65%CHO groups were further subdivided into 2 groups (*n* = 8) receiving either high-fibre LGI or lower-fibre HGI foods.	32	42	The German Ministry of Education and Research and the German Research Foundation
Lagerpusch, 2013b [50]	A controlled, nutritional, intervention study	Germany	Healthy men	Diets and formula meals		The study comprised 1 week of overfeeding (+50% of energy requirement), 3 weeks of energy restriction (−50% of energy requirement), and 2 weeks of refeeding (+50% of energy requirement). During refeeding, subjects were divided into two sub-groups receiving either high-fibre LGI (low-glycaemic index) or HGI (high-glycaemic index) foods.	16	42	The German Ministry of Education and Research and the German Research Foundation
Guebre-Egziabher, 2008 [51]	Crossover, intervention study	France	Healthy subjects	Dietary changes		Rapeseed oil was supplied with a daily intake of 20 mL and three fish meals per week (100 g of salmon, tuna, mackerel, herring, and sardines) which provided a mean of 1.25 g/day of EPA and DHA.	17 (10 males and 7 females)	70	Association de langue francaise pour l’etude du diabete et autre maladies metaboliques (ALFEDIAM-Servier) and Fondation pour la Recherche Medicale, France.
Le, 2009 [34]	Crossover design	Switzerland	Healthy males with and without a family history of type 2 diabetes	Isocaloric diet or the same isocaloric diet supplemented with fructose		Isocaloric diet = 55% CHO, 30% fat and 15% protein. Fructose supplement (+35% of energy requirements). The fructose provided was equally consumed as a 20% solution with the 3 main meals.	16	42 (7 days of study + 35 washout days)	Supported by grants from the Swiss National Science Foundation and by grants from the Novartis Foundation and Takeda.
Muller, 2015 [52]	Crossover study	Germany	Healthy males	Dietary intervention		50% of the energy intake was given as a liquid-formula diet. The remaining 50% of energy was provided as a high-glycaemic index and low-glycaemic index meals and snacks	42	32	The German Ministry of Education and Research, the German Research Foundation, and the BMBF Kompetenznetz Adipositas, Core Domain “Body composition”
Ryan, 2012 [35]	Longitudinal study	United States of America	Obese postmenopausal women with impaired glucose tolerance	Weight loss program		Participants were instructed to reduce their caloric intake by 500 kcal/day	95	180	The Baltimore Veterans Affairs Medical Research Service, a Veterans Affairs Research Career Scientist Award, the Department of Veterans Affairs and Veterans Affairs Medical Centre GRECC, National Institute on Aging Grants, Claude D. Pepper Older Americans Independence Centre Grant P30-AG-028747, the National Institute of Diabetes and Digestive and Kidney Diseases Mid-Atlantic Nutrition Obesity Research Centre, and the General Clinical Research Centre of the University of Maryland, Baltimore, Maryland.
Brøns, 2004 [53]	Randomised, double-blinded, crossover intervention study	Denmark	Overweight men with a genetic predisposition for type II diabetes mellitus	Taurine	Placebo	1.5 g	18	112 + 14 wash-out	Steno Diabetes Centre, Gentofte, Denmark and by Aase and Ejnar Danielsens Foundation, Lyngby, Denmark
Ryan, 2013 [54]	Randomised crossover study	Australia	Individuals with non-alcoholic fatty liver disease	Mediterranean Diet (MD)	Low fat-high carbohydrate diet (LF/HCD)	The MD high in monounsaturated fats (MUFA) olive oil and omega-3 (ω3PUFA). Total of 40% fat, 40% carbohydrate, and 20% protein. The LF/HCD 30% fat, 50% carbohydrate, and 20% protein	12 (6 men and 6 women)	84 + 42 wash-out	NHMRC Neil Hamilton Fairley Fellowship and an Early Career Researcher Grant from the University of Melbourne.

**Table 3 nutrients-14-02036-t003:** IVGTT and AIRg trials description.

Study	Study Design	Country	Subjects Characteristics	Dietary Treatment	Dietary Control	Dose	Sample Size	Duration (Days)	Funding Source
Alemzadeh, 1998 [55]	Randomised controlled trial	United States of America	Obese hyperinsulinaemic adults	Hypocaloric diet +Diazoxide	Hypocaloric diet + placebo	1260 kcal/day for females and 1570 kcal/day for males comprised of liquid shakes (160 kcal/packet) and bars (150 kcal/bar) for six days. On the seventh day, participants consumed a hypocaloric diet (Optimealplan). Diazoxide 2 mg/kg/day for 8 weeks.	20 females and 4 males	56	The American Heart Association.
Osterberg, 2015 [56]	Randomised, double-blind placebo-controlled study	United States of America	Healthy young male adults	High-fat and hypercaloric diet + prebiotic	High-fat and hypercaloric diet + placebo	High-fat diet (55% fat, 30% carbohydrate, and 15% protein). Two sachets of VSL#3 prebiotic (450 billion bacteria per sachet)	20	42	VSL Pharmaceuticals Inc.
Jans, 2012 [57]	Randomised controlled trial	Ireland, Netherlands, Norway, Sweden	Subjects with the metabolic syndrome	Three isoenergetic diets: high MUFA (HMUFA) or two low-fat, high complex carbohydrate diets supplemented with n-3 PUFA	High SFA (HSFA) with a control capsule	HSFA(38%E) = SFA-rich diet (16E% SFA, 12E% mono unsaturated fatty acids (MUFA), 6E% PUFA), HMUFA(38%E) = MUFA-rich diet (8E% SFA, 20E% MUFA, 6E% PUFA), LFHCC(28%E) = high-complex carbohydrate diet (8E% SFA, 11E% MUFA, 6E% PUFA), with a control capsule (1 g per day), LFHCCn-3 = high-complex carbohydrate diet (8E% SFA, 11E% MUFA, 6E% PUFA), with a long-chain n-3 PUFA supplement (1.24 g p/d of eicosapentaenoic and docosahexaenoic acid, ratio 1.4:1)	84 men and women	84	Dutch Diabetes Research Foundation, the Johan Throne Holst Foundation, and Freia Medical Foundation.
Ard, 2004 [36]	Randomised study	United States of America	Subjects with above optimal blood pressure through stage 1 hypertension	Established diet (group B) or established + DASH diet (group C)	Advice only (group A)	Group B: Participants received concealing on low sodium/fat diets, an aim of 2.4 g/day or less of sodium and 30% of calories from fat weight loss, moderate alcohol intake, and increased physical activity of at least 180 min per week. minutes a week. Daily goals for group C were similar to those of group B, except goals for 25% of calories from fat, with 7% of calories from saturated fat; 9–12 servings of fruits and vegetables per day; and 2–3 servings of low-fat dairy per day	52 men and women	183	*Not disclosed*
Davy, 2002 [58]	Randomised trial	United States of America	Healthy men	Oat cereal	Wheat cereal	Oat group: 60 g of Quaker Oatmeal and 76 g of Quaker oat bran ready-to-eat cold cereal. Wheat group: consumed 60 g of Mother’s whole-wheat hot natural cereal and 81 g of frosted mini wheats.	36 men, 18 per group	84	The Quaker Oats Company
Kolehmainen, 2012 [59]	Randomised controlled dietary intervention	Finland	Individuals with features of metabolic syndrome.	Fresh bilberries	Habitual diet	400 g of fresh bilberries	27 men and women	112	Tekes—the Finnish Funding Agency for Technology and Innovation, the Academy of Finland, the Nordic Centre of Excellence on Systems Biology in Controlled Dietary Interventions and Cohort Studies, the European Nutrigenomics Organisation, the Yrj¨o Jahnsson Foundation, the Juho Vainio Foundation, the ABS Graduate School, and the Medical Research Fund of Tam pere University Hospital.
Larson-Meyer, 2006 [37]	Randomised study	Australia	Overweight subjects	Participants were randomised into three groups: calorie restriction (CR), calorie restriction + energy expenditure through structured exercise (CREX), and weight loss by a low-calorie diet followed by weight maintenance for 6 months (LCD)	Control diet (100% requirements)	CR = 25% calorie restriction, CREX = 12.5% calorie restriction +12.5% energy expenditure through structured exercise, LCD = 15% weight loss by a low-calorie diet followed by weight maintenance for 6 months	46 men and women	180	The National Health and Medical Research Council of Australia
Tierney, 2011 [60]	Randomised dietary intervention study	Ireland, United Kingdom, France, Sweden, Poland, Netherlands, Spain and Norway	Subjects with metabolic syndrome	Participants were randomised to four different diets: high-fat SFA-rich diet (high SFA (HSFA), high-fat MUFA-rich diet (HMUFAs), isoenergetic low-fat, high complex carbohydrate diet (LFHCC), isoenergetic low-fat, high complex carbohydrate diet (LFHCC n-3)		HSFA = 38% energy from fat and SFA-rich diet (16% SFA, 12% MUFA, 6% PUFA). HMUFAs = 38% energy from fat, MUFA-rich diet (8% SFA, 20% MUFA, 6% PUFA). LFHCC = 28% energy from fat (8% SFA, 11% MUFA, 6% PUFA) with 1 g day high oleic sunflower oil supplement. LFHCC n-3 = 28% energy from fat (8% SFA, 11% MUFA, 6% PUFA) with 1.24 g day VLC n-3 PUFA supplement	417 (185 males and 232 females)	84	The EU 6 Framework Food Safety and Quality Programme, ‘Diet, genomics, and the metabolic syndrome: an integrated nutrition, agro-food, social and economic analyses. The Norwegian Foundation for Health and Rehabilitation, South-Eastern Norway Regional Health Authority, the Johan Throne Holst Foundation for Nutrition Research, and the Freia Medical Research Foundation.
Brady, 2004 [61]	2-period, parallel dietary intervention	United Kingdom	Indian Asians (males)	Participants were randomly assigned to consume the high (corn oil-based) n-6 PUFA cooking oils and spread them with their usual diet for the first six weeks. For the second 6-weeks of the study the participants consumed a daily supplement of n-3 LC-PUFA.		16 g of spread, 21 g from cooking oils per day. Daily supplement: 4.0 g of fish oil, 2.5 g of EPA + DHA	29	84	Food Standards Agency of the United Kingdom
Fava, 2013 [62]	Randomised, controlled, single-blind, parallel design	United Kingdom	Men and women, aged between 30 and 65 years, with normal hepatic and renal function.	Participants followed a 4-week run-in reference diet that was a high saturated fat diet (HS; saturated fatty acids, SFA)- high glycemic index (GI) diet (38% fat), after which they were randomly assigned to either continue with the reference diet or one of four experimental diets HM/LGI, HC/HGI, HM/HGI or HC/LGI	HS: total fat 38%E, SFA 18%E, MUFA 12%E, PUFA 6%E, CHO 45%E, GI 64%.	HM/HGI: total fat 38%E, SFA 10%E, MUFA 20%E, PUFA 6%E, CHO 45%E, GI 64%; HM/LGI: total fat 38%E, SFA 10%E, MUFA 20%E, PUFA 6%E, CHO 45%E, GI 53%; HC/HGI: total fat 28%E, SFA 10%E, MUFA 11%E, PUFA 6%E, CHO 55%E, GI 64%; HC/LGI: total fat 28%E, SFA 10%E, MUFA 11E, PUFA 6%E, CHO 55%E, GI 51%.	88, 43 men and 45 women	196	UK Food Standards Agency
Giacco, 2013 [63]	Randomised, controlled, parallel-group design	Italy and Finland	Healthy subjects	Diet based on wholegrain	Refined cereals	The wholegrain diet in Naples (Italy): wholegrain products include whole wheat bread (plus some endosperm rye bread), whole wheat pasta, barley kernels, wholegrain oat biscuits, and breakfast cereals (all bran sticks and flakes). Participants in Kuopio (Finland) were advised to replace their habitual potato consumption with 210 g dry weight of whole wheat pasta per week and were given whole oat biscuits for snacks.	123 men and women	84	European Commission in the 6th Framework Programme, Project HEALTHGRAIN, by Raisio Plc Research Foundation (JL), the Nordic Centre of Excellence projects “HELGA whole grains and health”, “SYSDIET Systems biology in controlled dietary interventions and cohort studies” (MK, US, MU). Barilla G&R F.lli. SpA, Parma, Italy and Raiso Nutrition Ltd., Finland.
Juntunen, 2003 [64]	Randomised crossover trial	Finland	Healthy postmenopausal women.	Participants consumed high-fibre rye bread and white-wheat bread. Participants acted as their controls.		One portion of rye bread contained on average 206 kJ and 4.4 g of fibre. One portion of wheat bread contained 241 kJ and 0.6 g of fibre. A minimum of 4–5 portions of the test bread had to be eaten each day, and the number of portions to be eaten varied according to the daily energy intake of the individual.	20	56 + 56 wash-out days	Not disclosed
Kien, 2013 [40]	Two-treatment, Two-period, two-sequence crossover design.	United States of America	Young adults.	All subjects ingested a low-fat/low-PA (palmitic acid), baseline control diet for 7 days. Then, subjects participated in a crossover study of two 3-week experimental diets. One diet was designed to resemble the habitual diet and was high in PA (HPA) or a diet low in PA and high in OA (oleic acid) (HOA)		HPA = 40.4% kcal; PA, 16.0% kcal; and OA, 16.2% kcal. HOA = 40.1% kcal; 2.4% kcal; and 28.8% kcal, respectively	18 men and women	28 + 7 wash-out days	National Institutes of Health Grants, and the National Centre for Research Resources, National Institutes of Health, U.S. Public Health Service.
Douglas, 2006 [65]	Crossover design	United States of America	Women with polycystic ovary syndrome	Low carbohydrate diet (Low CHO), MUFA diet (monounsaturated fatty acid), and STD diet (standard diet)	The CHO diet comprised 2014 calories, 43% of CHO, 45% of fat, and 15% of proteins. The MUFA diet comprised 2006 calories, 55% of CHO, 15% of proteins, and 33% of fats	STD diet comprised 56% CHO, 31% fat and 16% protein, and 2000 calories	11	48 + 42 wash-out days	Not disclosed
Paniagua, 2007 [38]	Crossover design	Spain	Insulin resistant subjects	Participants consumed three different diets: (1) diet enriched in saturated fat (SAT), (2) diet rich in monounsaturated fat (MUFA), and (3) diet rich in carbohydrates (CHOs)		CHO diet contained 65% CHO and 20% fat (6%SAT, 8% MUFA, and 6% polyunsaturated fat (PUFA)). The MUFA diet contained 47% CHO and 38% fat (9% SAT, 23% MUFA, 75% of which was provided as an extra virgin olive oil, and 6% PUFA), the rich diet. The SAT diet contained 47% CHO, 15% protein, and 38% fat (23% SAT, 9% MUFA, and 6% MUFA)	11, 4 men and 7 women	84	The Spanish Arteriosclerosis Foundation, the Pharmaceutics Foundation AVEN ZOAR of Seville (2004); the Medical College of Cordova Foundation (2004); and the Secretaria General de Calidad y Eficiencia, Junta de Andalucia (78/02 and 240/04).
Davis, 2012 [39]	Randomised trial	United States of America	Overweight minority adolescents.	Nutrition newsletter	Strength newsletter	The nutrition newsletter covered tips on how to continue to eat foods and drink beverages low in sugar and high in fibre and included one or two new low-sugar or high-fibre recipes. The strength training newsletter covered the benefits of that type of exercise and sample strength-training exercises.	53 adolescents, 24 males, and 29 females.	240	The National Institutes of Cancer, University of Southern California Centre for Transdisciplinary Research on Energetics and Cancer, the National Institute of Child Health and Human Development, the National Cancer Institute (Cancer Control and Epidemiology Research Training Grant), the Dr Robert C. and Veronica Atkins Foundation
Davis, 2009 [66]	Randomised controlled trial	United States of America	Overweight Latino adolescents	Nutrition intervention group and Nutrition intervention + strength training group	Control	The dietary intervention targeted two goals: <10% of total daily calorie intake from added sugar and consuming at least 14 g/1000 kcal of dietary fibre a day. The nutrition education + strength training, participants received strength training twice per week (60 min/session). The control group followed their usual diet	54. Control = 16, nutrition education = 21, nutrition + strength training = 17	112	The National Institutes of Cancer, University of Southern California Centre for Transdisciplinary Research on Energetics and Cancer, the National Institute of Child Health and Human Development, the National Cancer Institute (Cancer Control and Epidemiology Research Training Grant), the Dr Robert C. and Veronica Atkins Foundation

**Table 4 nutrients-14-02036-t004:** Lin’s concordance correlation coefficient (LCCC) between gold standards and surrogate markers using proportional change data.

	LCCC	95% Confidence Intervals	
AIRg and DI	0.14	−0.11	0.75
Clamp and fasting insulin	0.76	−0.33	0.24
Clamp and HOMA-IR	0.17	−0.76	0.13
Clamp and AUC OGTT	0.29	−0.33	1.1
IVGTT (Si) and fasting insulin	0.15	−0.37	0.06
IVGTT (Si) and HOMA-IR	0	−1.13	−0.42
IVGTT (Si) and AUC OGTT	0.93	−0.35	0.38

AIRg = Acute Insulin Response to glucose, DI = disposition index, HOMA-IR = Homeostatic Model Assessment for Insulin Resistance, AUC OGTT = Area Under the Curve Oral Glucose Tolerance Test, LCCC = Concordance Correlation Coefficient. An LCCC of <0.99 is considered poor concordance.

**Figure 2 nutrients-14-02036-f002:**
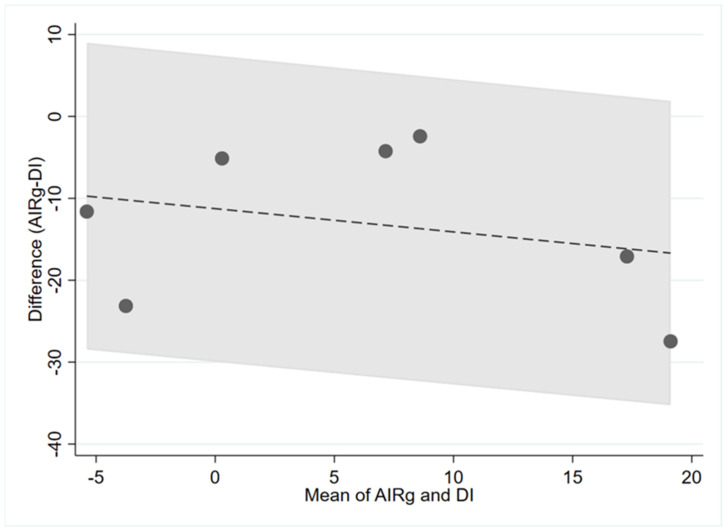
Bland–Altman plots between of AIRg vs. DI (insulin secretion). The central dashed line represents the mean difference between measures represented as log values. The area in grey represents the 95% confidence intervals. AIRg = acute insulin response to glucose, DI = disposition index.

**Figure 3 nutrients-14-02036-f003:**
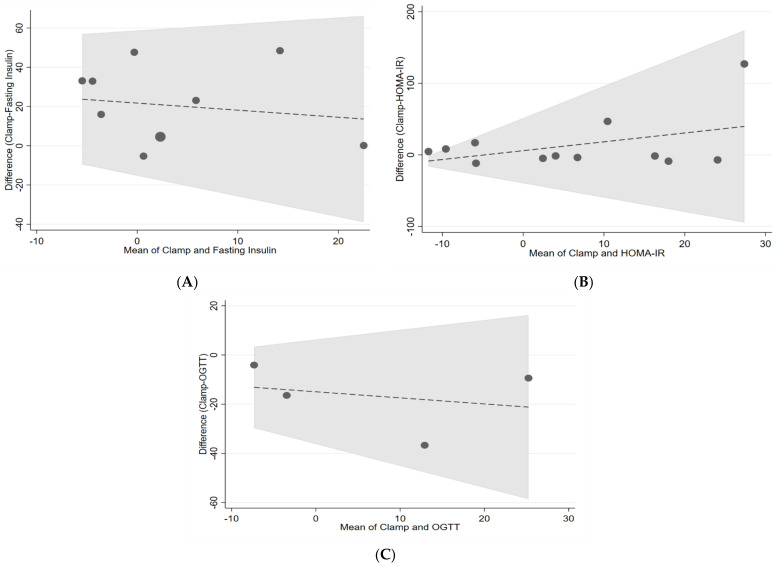
Bland–Altman plots of (**A**) clamp vs. fasting insulin, (**B**) clamp vs. HOMA-IR, (**C**) clamp vs. OGTT. The central dashed line represents the mean difference between measures represented as log values. The area in grey represents the 95% confidence intervals. HOMA-IR = homeostatic model assessment for insulin resistance, OGTT = oral glucose tolerance test.

**Figure 4 nutrients-14-02036-f004:**
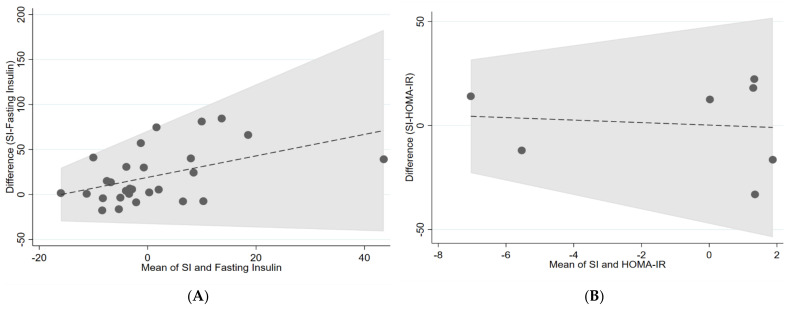
Bland–Altman plots of (**A**) SI vs. fasting insulin, (**B**) SI vs. HOMA-IR, (**C**) SI vs. OGTT. The central dashed line represents the mean difference between measures represented as log values. The area in grey represents the 95% confidence intervals. HOMA-IR = homeostatic model assessment for insulin resistance, OGTT = oral glucose tolerance test.

## 4. Discussion

The worldwide rapidly increasing prevalence and burden of type 2 diabetes amplifies the need and urgency for testing and innovative strategies to prevent the development of the disease.

Earlier identification and acceptance of efficacy markers for diabetes risk (or the recognition of gaps in evidence needed for their acceptance) will lead to an earlier and wider endorsement of their use and can speed the testing and development of potentially promising strategies to counteract the disease.

Dietary interventions are modifiable factors that may be more effective than pharmacological agents in preventing the onset of diabetes [67]. Furthermore, the dietary composition may influence insulin sensitivity and secretion [68,69,70].

In this systematic literature review, we identified studies using surrogate markers of insulin sensitivity and secretion and its correlation with gold standards in a population without diabetes using dietary interventions. The use of surrogate markers for the assessment of insulin sensitivity and secretion has been proposed to evaluate large population-based epidemiological investigations to save costs and to obtain data in quick time.

Nonetheless, even though the use of surrogate markers to assess insulin sensitivity and secretion have the potential to decrease costs and minimise participant burden and risk [17], their utility appears to be limited, at least in this review.

One problem that some surrogate markers have, more specifically fasting insulin, HOMA-IR, and AUC OGTT, is that they can be unreliable in certain populations such as the elderly and those with uncontrolled diabetes [20,71]. In this review, the study made by Hayes (2006) used elderly subjects.

Another fasting insulin limitation is that it lacks standardisation of the insulin essay procedure [17]. AUC OGTT provides information regarding glucose tolerance but not insulin resistance [17]. Whereas HOMA-IR and disposition index, evaluate hepatic insulin resistance more than peripheral insulin sensitivity which was our target [20,72].

Our study differs from previously published data, which found a correlation between QUICKI or the OGGT-based indices (Stumvoll, OGIS, Matsuda, and Gutt) with the hyperinsulinaemic-euglycaemic clamp to assess insulin sensitivity [12]. However, the investigators of that study analysed their data using a correlation coefficient method. This method is considered an incorrect measure of reproducibility or repeatability as it measures the correlation between variables but not the agreement between them [73].

The evaluation of the trials involved in our research used different units for the surrogate markers. Thus, the different evaluation parameters made it challenging to integrate the outcomes. However, we decided that the best way to compare the effectiveness of the surrogate markers with the gold standards was with a proportional change formula, analysed using Lin’s concordance correlation coefficient, which is also an index of reliability.

We expected to see an agreement between the surrogate measures and the gold standards with the Bland–Altman plots; however, this was not the case.

Advantages and Disadvantages

One of the main strengths of this study was that the population analysed was diverse, as it included all sexes and ages and some of the participants presented a condition that can lead to type 2 diabetes or any other cardiovascular disease. This study was a general comparison instead of focusing on a specific population.

On the contrary, some might consider this a disadvantage because the analysed participants were not a heterogenic population. Two studies conducted on adolescents were included in this review [39,66].

Another strength was the meticulous selection strategy that identified all the accessible trials with our requested characteristics. So far, no other systematic review or meta-analysis has examined the effect of dietary treatment on insulin sensitivity and surrogate markers.

All the studies included in this review are high-quality trials according to the risk assessment. Nonetheless, some studies showed no information regarding allocation concealment and random sequence generation. This, however, did not influence our results.

## 5. Conclusions

There is insufficient evidence to evaluate the effects of nutritional intervention studies on insulin sensitivity or secretion with surrogate markers when compared to a euglycemic-hyperinsulinaemic clamp, IVGTT or AIRg.

Future research is needed to investigate if surrogate measures of insulin sensitivity and secretion can be repeatable and reproducible in the same way as the gold standards using nutritional intervention studies.

## Figures and Tables

**Figure 1 nutrients-14-02036-f001:**
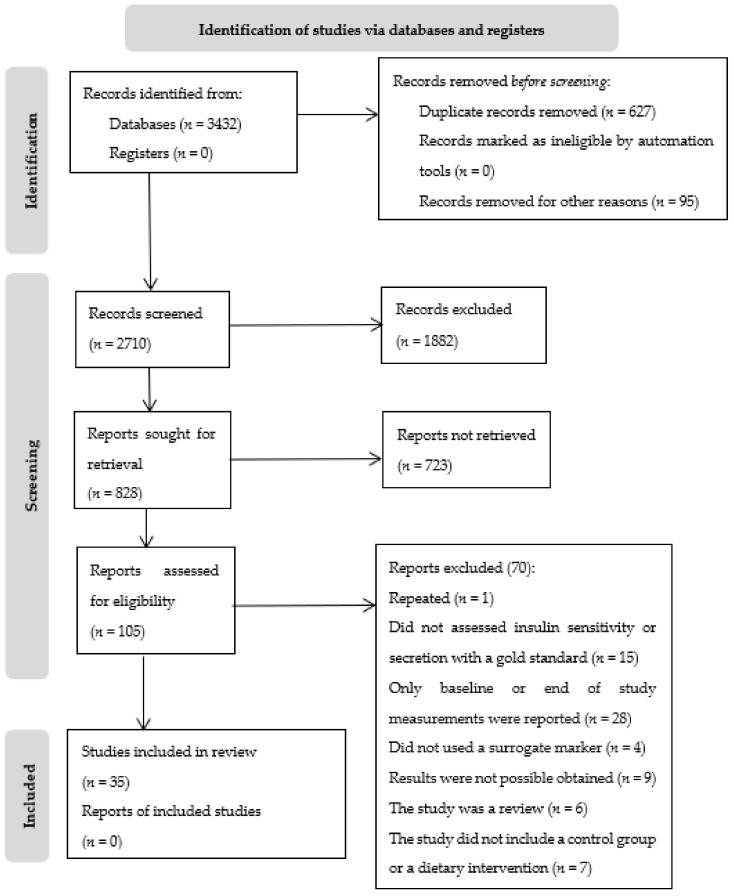
PRISMA flow diagram of study selection for systematic review.

**Table 1 nutrients-14-02036-t001:** Search strings.

Number	Search String
1	(insulin-secreting cells[MeSH Terms] OR insulin secretion[Title/Abstract] OR intravenous glucose tolerance[Title/Abstract] OR glucose tolerance test[MeSH Terms] OR hyperglycaemic clamp[Title/Abstract] OR hyperglycemic clamp[Title/Abstract] OR insulin resistance[MeSH Terms] OR insulin resistance[Title/Abstract] OR glucose clamp technique[MeSH Terms])
2	(diet[MeSH Terms] OR diet[Title/Abstract] OR dietary[title/abstract] OR food[MeSH Terms] OR nutrition[MeSH Terms]) AND (intervention[Title/Abstract] OR trial[Title/Abstract])

## Data Availability

The datasets generated and/or analysed during the current study have been made available online.

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
