# Peer review of "The Use and Effectiveness of Selected Alternative Markers for Insulin Sensitivity and Secretion Compared with Gold Standard Markers in Dietary Intervention Studies in Individuals without Diabetes: Results of a Systematic Review"

_nutrients, 2022, doi:10.3390/nu14102036_

Round 1

Reviewer 1 Report

This article is a review to identify if surrogate markers of insulin sensitivity and secretion in response to dietary intervention could be a better new gold-standard methodology to assess insulin sensitivity and insulin secretion. The test to determine insulin sensitivity is the euglycemic-hyperinsulinemic clamp (EHC). Another assessment of insulin sensitivity is the intravenous glucose tolerance test (IVGTT). For insulin secretion, the method is the hyperglycemic clamp (HC) and the first phase of insulin secretion of IVGTT or Acute Insulin Response to Glucose (AIRg). All these methods imply problems for the patients. Therefore there are surrogate markers, which provide indirect estimates; for example insulin (or C-peptide) and glucose measurements in blood samples taken in the fasting state or after a glucose challenge or a standardized meal.

They worked with many articles for comparisons. The tables are almost impossible to read because they are too long. I suggest breaking the tables according to the methods used or classifying them in other way. 

They conclude that there is not enough evidence to predict insulin sensitivity and secretion with surrogate markers when compared to gold standards in nutritional intervention studies, and more research is needed. The authors should discuss more why the principal surrogate methods do not give enough information.

Reviewer 2 Report

This well-performed systematic review examines the most used markers for both insulin resistance and insulin secretion. I do have some comments:

  • Only 2 databases have been searched, while normally about 4-5 are more "standard" (e.g. Embase, Scopus,…). The authors should state why this is the case.
  • Line 293-297: the description of Lin’s concordance correlation coefficient would better fit the Materials and Methods section (which is already mostly done), in my view.
